# Arachidonic Acid Metabolism and Kidney Inflammation

**DOI:** 10.3390/ijms20153683

**Published:** 2019-07-27

**Authors:** Tianqi Wang, Xianjun Fu, Qingfa Chen, Jayanta Kumar Patra, Dongdong Wang, Zhenguo Wang, Zhibo Gai

**Affiliations:** 1Traditional Chinese Medicine History and Literature, Institute for Literature and Culture of Chinese Medicine, Shandong University of Traditional Chinese Medicine, Jinan 250355, China; 2Institute for Literature and Culture of Chinese Medicine, Shandong University of Traditional Chinese Medicine, Jinan 250355, China; 3Key Laboratory of Traditional Chinese Medicine for Classical Theory, Ministry of Education, Shandong University of Traditional Chinese Medicine, Jinan 250355, China; 4The Institute for Tissue Engineering and Regenerative Medicine, The Liaocheng University, Liaocheng 252000, China; 5Research Institute of Biotechnology & Medical Converged Science, Dongguk University-Seoul, Goyangsi 10326, Korea; 6Institute of Clinical Chemistry, University Hospital Zurich, University of Zurich, Wagistrasse 14, 8952 Schlieren, Switzerland; 7Guizhou University of Traditional Chinese Medicine, Fei Shan Jie 32, Guiyang 550003, China

**Keywords:** arachidonic acid, cyclooxygenase, lipoxygenase, cytochrome P450, kidney inflammation, therapeutic target

## Abstract

As a major component of cell membrane lipids, Arachidonic acid (AA), being a major component of the cell membrane lipid content, is mainly metabolized by three kinds of enzymes: cyclooxygenase (COX), lipoxygenase (LOX), and cytochrome P450 (CYP450) enzymes. Based on these three metabolic pathways, AA could be converted into various metabolites that trigger different inflammatory responses. In the kidney, prostaglandins (PG), thromboxane (Tx), leukotrienes (LTs) and hydroxyeicosatetraenoic acids (HETEs) are the major metabolites generated from AA. An increased level of prostaglandins (PGs), TxA_2_ and leukotriene B4 (LTB_4_) results in inflammatory damage to the kidney. Moreover, the LTB_4_-leukotriene B4 receptor 1 (BLT1) axis participates in the acute kidney injury via mediating the recruitment of renal neutrophils. In addition, AA can regulate renal ion transport through 19-hydroxystilbenetetraenoic acid (19-HETE) and 20-HETE, both of which are produced by cytochrome P450 monooxygenase. Epoxyeicosatrienoic acids (EETs) generated by the CYP450 enzyme also plays a paramount role in the kidney damage during the inflammation process. For example, 14 and 15-EET mitigated ischemia/reperfusion-caused renal tubular epithelial cell damage. Many drug candidates that target the AA metabolism pathways are being developed to treat kidney inflammation. These observations support an extraordinary interest in a wide range of studies on drug interventions aiming to control AA metabolism and kidney inflammation.

## 1. Introduction

Arachidonic acid (AA), also named eicosa-5,8,11,14-tetraenoic acid, is a ω-6 polyunsaturated fatty acid (PFA) and is mainly present in the form of phospholipids in the cell membrane. When cells are under stress, AA is released from the phospholipids by phospholipase A_2_ (PLA_2_) and phospholipase C (PLC) as free arachidonic acids [1,2,3], which become the precursor of proinflammatory bioactive mediators through three metabolic pathways. Through the cyclooxygenase (COX) pathway, AA can be metabolized into prostaglandins (PGs) and thromboxanes (TXs). AA can also be converted into leukotrienes (LTs) and lipoxins (LXs) by the lipoxygenase (LOX) pathway [4,5,6]. Moreover, AA also generates epoxyeicosatrienoic acids (EETs) or hydroxyeicosatetraenoic acids (HETEs) through the cytochrome P450 (CYP450) pathway. Together, these AA metabolites are referred as eicosanoids, which are effective autocrine and paracrine bioactive mediators, and are widely involved in a variety of physiological and pathological processes [7,8,9,10].

Kidney inflammation, characterized by hematuria, proteinuria, edema, hypertension, etc., is caused by immune-mediated inflammatory mediators (such as complement, cytokines, reactive oxygen species, etc.), resulting in a group of kidney diseases with a varying degree of renal dysfunction. Without prompt treatment, it will lead to thromboembolism, acute renal failure (and even chronic nephritis), chronic renal failure and finally uremia [11].

The relationship between AA and inflammation attracts our interest in the effect of AA metabolism on kidney inflammation. Therefore, we systematically summarized the effect of AA-derived bioactive mediators on kidney inflammation by discussing the regulatory mechanism of AA metabolism in the kidney, followed by the mechanism of AA-induced renal inflammation and a potential treatment targeting the AA metabolism.

## 2. Regulation of the AA Metabolism in the Kidney

### 2.1. The Release of AA

Normally, AA exists in the cell membrane in the form of phospholipids. When the cell membrane is subjected to stimuli, especially the inflammatory reaction, the phospholipids are released from the cell membrane. Through the hydrolysis of phospholipids by PLA_2_ and PLC [1,3,12], AA is released and then transformed into a bioactive metabolite with the help of different enzymes, thus promoting inflammatory cascades. At present, it is well known that at least three metabolic pathways (the COX pathway, LOX pathway and CYP450 pathway) are involved in the metabolism of AA, which are closely related to the occurrence, development, and regression of renal inflammation (Figure 1) [1].

### 2.2. COX Pathway

COX-1/2, also called prostaglandin H synthase (PGHS), is one of the key enzymes involved in the AA metabolism [13]. The constitutive enzyme COX-1 is responsible for the expression of prostaglandin E_2_ (PGE_2_) at the background level or the expression of PG under hypotonic swelling stimulation [14]. COX-2 is almost not expressed at normal physiological conditions; however, it is highly expressed when the kidney is under the influence of stimuli, such as chronic sodium deficiency and ultrafiltration [15]. It is important to note that COX-2 is the dominating source of prostacyclin [16]. Once AA is released, it can be metabolized by two isozymes of PGHS, PGHS-1 and PGHS-2 [17]. PGHSs have two different but complementary enzyme activities, one is cyclooxygenase (dioxygenase) activity, which catalyzes the production of PGG_2_ from arachidonic acid, and the other is peroxidase activity, which promotes the reduction of PGG_2_ to PGH_2_ [18]. In general, the COX protein contains a cyclooxygenase site and a peroxidase site. AA is converted to the endogenous hydrogen peroxide PGG_2_ by the cyclooxygenase site, and the peroxidase site is responsible for reducing PGG_2_ to PGH_2_ [19]. COXs initially metabolize AA to unstable PGG_2_ by their COX function and then convert it to PGH_2_ by their peroxidase function. PGH_2_, like cell-and tissue-selective prostanoid synthases and isomerases, is not stable, and can easily generate many bioactive prostaglandins, such as prostaglandins D2, E2, F2α, and I_2_, and TXA_2_, depending on the differential expression of these synthetases in different tissues [20,21,22,23]. PGH_2_ can also be decomposed into malonaldehyde (MDA) and 12L-hydroxy-5,8,10-heptadecatrienoic acid (HHT) by thromboxane synthase [24]. The process of converting PGH_2_ to PGD requires two PGD synthetases, namely hematopoietic PGD synthase (H-PGDS) and lipocalin-type PGD synthase (L-PGDS). Then the PGD_2_ is metabolized to 15-deoxy-Δ 12,14-prostaglandin J 2 (15d-PGJ 2) [25]. PGI_2_ is extremely unstable, with a half-life of only two to three minutes, and is easily converted to 6-keto-prostaglandin F1α spontaneously [26,27]. In particular, PGH_2_ is produced in most cells by the action of microsomal PGE_2_ synthase (mPGES). PGE_2_ is synthesized in almost all human cells and exerts extremely complex physiological effects in the inflammatory response through the signaling pathway which is composed of four G protein-coupled receptors: E-type prostanoid receptors (EP)1, EP2, EP3 and EP4 [28,29,30]. As for TXA_2_, AA can generate TXA_2_ via the action of thromboxane synthase [31], which has a half-life of only about 30 s and is then spontaneously converted to thromboxane B_2_ (TXB_2_) [32,33]. Three things that need to be especially pointed out are that platelets mainly form TXA_2_, endothelial cells mainly form PGI_2_, and PGE_2_ is the main prostatic body produced by renal collecting tubule cells [34]. In recent years, another variant of the COX family, COX-3 (or CX-1b), which is an allosteric splice variant of COX-1, has been discovered, and its gene sequence differs from COX-2 in such a way that the COX-3 retains the intron 1 sequences [35]. COX-3 also catalyzes the production of PGH_2_, its activity can be inhibited by crude aminophenol, and it is mainly expressed in the microvessels of the brain and heart [36,37,38]. Hence, its role in AA metabolism and kidney inflammation needs to be further explored.

### 2.3. LOX Pathway

Under the catalysis of lipoxygenase, AA is metabolized into hydroperoxyeicosatetraenoic acid (HpETE). Current research suggests that at least four enzymes, namely 5-LOX, 8-LOX, 12-LOX, and 15-LOX are involved in the metabolism of AA in the LOX pathway. However, the 5-, 12-, and 15-positions are the main oxidation sites of AA, leading to oxidation reactions that are based on the catalysis of 5-LOX, 12-LOX, and 15-LOX enzymes. In this review, we focus on the 5-LOX, 12-LOX and 15-LOX pathways (Figure 2).

Human 5-LOX holds a major function in kidney inflammation, ranging from kidney tubules to glomeruli. In this pathway, AA forms 5-hydroperoxyeicosatetraenoic acid (5-HpETE) by dioxygenase [39], and then 95% is generated into 5-hydroxyeicosatetraenoic acid (5-hydroxy-6,8,11,15-eicosatetraenoic acid, 5-HETE) at C7 and 5% is generated into 8-HETE at C10, which are the first two steps during the conversion process of AA to proinflammatory LTs [40]. Besides, Oxo-ETE is generated via the LOX product, HETEs by the microsomal dehydrogenase in the human polymorphonuclear leukocytes (PMNLs) [39,41]. Then, oxo-ETEs are formed through the oxidation of HETEs [42,43], which is the strongest eosinophil chemoattractant among bioactive mediators. From here, AA metabolites are further converted to LTA_4_ and LXs via 5-LOX activator protein (5-FLAP) and dehydrase [44]. Basically, the catalytic function of the 5-LOX enzyme is mainly manifested in the following two aspects: one is the insertion of molecular oxygen by dioxygenase activity and the other is the formation of epoxide by LTA_4_ synthase activity. Regarding 5-HpETE, it is generated from AA through homolytic cleavage and removal of hydrogen on the pro-S hydrogen at carbon-7 [39]. 5-LOX can generate LTA_4_ by removing the C10 hydrogen atom from 5-HpETE [45]. LTA_4_ is unstable and can be hydrolyzed or combined with glutathione or transcellularly transferred to generate bioactive eicosanoids. In neutrophils and other inflammatory cells of kidney tissue, LTA_4_ is catalyzed by epoxide hydrolase to form LTB_4_ [46]. LTA_4_ and glutathione (GSH) catalyze the production of LTC_4_ using glutathione S-transferase (GST) and then remove glutamic acid to form LTD_4_ via γ-glutamyl transferase, which is further metabolized by dipeptidase to form LTE_4_. Then, LTF_4_ is synthesized from LTE_4_ by γ-glutamyl transferase [47].

The roles of the 12-LOX and 15-LOX pathways are mainly in the production of HETEs and LXs. The 12-LOX pathway is similar to the 5-LOX pathway. AA first generates 12-HpETE and 12-hydroxyeicosatetraenoic acid (12-HETE) via 12-LOX [48,49]. But there is a difference between them, as in addition to the conversion of AA to 12-HETE, 12-LOX can also convert 5(S)-HETE to 5(S),12(S)-dihydroxyeicosatetraenoic acid (-diHETE) as well as metabolize 15(S)-HETE to 14(R),15(S)-diHETE in the leukocytes. These products ultimately convert into extra-platelet LTA_4_ [50,51]. Another biosynthetic pathway for LXs involves 5-LOX in neutrophils and 12-LOX in the platelets. 5-LOX generates LTA_4_ in the neutrophil, which is then transferred to the platelet, where 12-LOX subsequently generates either LXA_4_ or LXB_4_ [45,52,53]. Same as the 12-LOX pathway, the principal effect of the 15-LOX pathway is to ultimately generate HETEs, ETEs and LXs. There are two isoforms in mammalian cells: 15-LOX-1 and 15-LOX-2. 15-LOX-1 (12/15-LOX), which is encoded by the arachidonate 15-lipoxygenase (ALOX15) gene, could metabolize AA into LXA_4_, LXB_4_ and 15-oxo-ETEs [43,45], while 15-LOX-2 will metabolize AA into 15-oxo-ETE and 8SHETE. HpETEs. In particular, when using 5, 15-diHpETE as a substrate, the primary product catalyzed by 12-LOX and 15-LOX-1 is LXB_4_, and the efficiency of 15-LOX-1 is 20 times higher than that of 12-LOX [45]. In humans, 15-LOX-1 and leukocyte 12-LOX have high homology and can form 12(S)-HETE and 15(S)-HETE simultaneously, so these two pathways can be collectively called 12/15-LOX (12/15-LOX). When 15-LOX-2 metabolizes AA to produce only 15-HETE, 15-LOX-1 also metabolizes linoleic acid to synthesize hydroxyoctadecadienoic acid [54]. Then, 15-HETE is rapidly converted into LXA_4_ or LXB_4_ by hydrolase [55].

### 2.4. Cytochrome P450 (CYP450) Pathway

The CYP450 pathway is the major metabolic pathway of AA in the kidney [56]. CYP450 can be detected in the endoplasmic reticulum, mitochondria and nuclear membrane of the kidney [57,58]. In general, AA produces corresponding metabolites mainly through three kinds of reduced triphosphopyridine nucleotide (NADPH, or reduced coenzyme II)-dependent oxidation in the CYP450 pathway. The first is the formation of 5,6-; 8,9-; 11,12-; and 14,15-epoxyeicosatrienoic acids (EETs) through surface oxidation [59,60,61,62,63]. The kidney is an organ with high expoxygenase activity, which, like the liver, also produces enantioselective EETs. In the case of lipoxygenase and cyclooxygenase, AA is first converted to hydrogen peroxide (HpETEs) [43], and then through CYP450 isozymes, oxo-ETEs can also be formed directly from HpETEs (just like the common process of forming HETEs via enzymatic oxidation) [42]. 14,15-EET is the main epoxy compound formed by kidney, and most of the isomers in this region are in (R,S) configuration. CYP2 is the main CYP450 epoxygenase family, and more importantly, CYP2C8 is the most paramount and plays a central epoxygenase role in the metabolism of AA to biologically active EETs in human kidney [64,65]. In addition, the CYP2J family also contributes to the formation of EETs in human and mouse kidneys [64,65]. Early research of CYP2C epoxygenases suggested that CYP2C29 and CYP2C39 produced 14,15-EET, and the CYP2C38 produced 11,12-EET [66]. In particularly, human CYP2J2 isoforms are extremely efficient at epoxidation of AA at the 14,15-position [67]. CYP2J5 is an epoxygenase enzyme that is confined in tubular cells and metabolizes AA to 8,9-EET; 11,12-EET; and 14,15-EET in the mouse kidney [68]. Therefore, different vascular smooth muscle cells and epithelial transport processes controlled by EETs have led to the differential localization and regulation of renal vascular and tubular CYP450 cyclooxygenase, and these conclusions are consistent throughout [69]. Then, EETs are mainly hydrolyzed by soluble epoxide hydrolase (sEH) to form 5,6-; 8,9-; 11,12-; and 14,15-dihydroxyepoxyeicosatrienoic acids (DHETs), which possess weak biological activity [70]. Recent research has confirmed that sEH is one of the key enzymes in the metabolism of EETs, and the regulation of sEH activity and can change the level of EETs in vivo [70].

The other two CYP450 pathways ultimately produce HETEs. One is the formation of 5-, 8-, 9-, 11-, 12-, or 15-HETEs by propylene oxidation. HpETEs produced by lipoxygenase and cyclooxygenase can be reduced to monohydroxy fatty acids (HETEs) by peroxidase [43], which means the hydroxyl group is adjacent to a conjugated diene system. CYP2J5 can also facilitate the process, which has been confirmed in tubular cells, and metabolizes AA to 11-HETE and 15-HETE in mouse kidney [68]. The other is the formation of 19- and 20-Hydroxyeicosatetraenoic (19- and 20-HETE) by ω-1 hydroxylation. The CYP ω-hydroxylase that has been discovered so far is mainly in the CYP4 family. In kidney, proximal straight tubules are capable of converting AA to 20-HETE and 19(S)-HETE by ω-hydroxylase [71]. What’s more, 20-HETE is the main product of AA catalyzed by CYP ω-hydroxylase, and in the human kidney, 20-HETE formation is mediated by both CYP4F2 and CYP4A11 [56,72]. Then, 20-HETE can be further oxidized to 20-carboxy arachidonic acid (20-COOH-AA) by alcohol dehydrogenase [73]. It is now well established from a few studies that, CYP4A1, CYP4A2, and CYP4A3 have remarkable AA ω-hydroxylase activity [74]. Moreover, CYP4A1 as an arachidonate ω-hydroxylase has a higher catalytic efficiency, with a turnover rate 20 times higher than that of CYP4A2 or CYP4A3 [74]. Besides, CYP4A1 can also convert AA to 11,12-EET in the mouse kidney [74].

## 3. Mechanism of AA-induced Renal Inflammation

### 3.1. PGs and Renal Inflammation

PGs produced by the COX pathway, function differently in case of renal vascular disease. The impact of PGs on renal inflammation is mainly focused in the field of lupus nephritis (LN). In mice and humans, urinary prostaglandin D synthase (uPGDS) is considered as a biomarker of LN, which makes the role of PGD_2_ important in the development of LN [75]. PGD_2_ are considered as an inflammatory marker, while lipocalin-like-prostaglandin-D synthase (L-PGDS) is considered as a urinary biomarker for human active lupus nephritis [76]. One study of 184 longitudinal observations in 80 patients showed that lipocalin-like-prostaglandin-D synthase could predict the onset/remission of LN [77]. In patients with systemic lupus erythematosus (SLE), increased expression of PGD_2_ receptors (PTGDRs) in blood basophils causes an increase of PGD_2_ metabolites in the plasma [78]. PGD_2_ regulates the inflammatory response through two receptors, PTGDR-1 and PTGDR-2, which are also called D prostanoid receptor-1 (DP-1) and D prostanoid receptor-2 (DP-2). DP-2 also functions as the chemo-attractant receptor-homologous molecule, which is expressed on the T helper type 2 (TH2) cells (CRTH2) [79,80]. Under in vivo condition, the targeted cells of PGD_2_ are mainly basophils, and the two PTGDRs are expressed at the highest level in the peripheral blood leukocytes. Interestingly, PTGDR-1 is ubiquitously expressed in the leukocytes, whereas PTGDR-2 only mediates the activation and chemotaxis of basophils, eosinophils, and CD4^+^ TH2 cells [81]. PGD_2_ and PTGDR induce the activation and infiltration of basophils in the kidneys of patients with SLE via mediating C-X-C motif ligand 12 (CXCL12) [78]. CXCL12 plays a biological role mainly through the chemokine receptor CXCR4, and regulates the physiological distribution of neutrophils in the kidney [82,83]. Inflammatory tissues and secondary lymphoid organs (SLOs) are associated with high levels of CXCL12, and CXCL12 mediates immune cell recruitment, which are the factors associated with the pathogenesis of SLE [78]. More importantly, basophil autocrine PGD_2_ is the main factor of PGD2-induced CXCR4 epoxidation. In humans and mice, the CXCL12-CXCR4 axis promotes the accumulation of basophils in SLOs during lupus through its mediating effect on the PGD_2_-PTGDR axis [78]. At the same time, we also can’t deny the bad consequence caused by PGE_2_ and TXB_2_ in the LN. High levels of PGE_2_ and TXB_2_ promoted the activation of T cells and the production of IL-4 and IL-10 cytokines, which in turn leads to glomerulosclerosis in LN [84].

In addition, a potential link between PGs and tubulointerstitial lesions and glomerulonephritis was well studied. In the kidney with unilateral ureteral obstruction (UUO), PGD_2_ has penetrated all the sides of tubulointerstitial lesions by DP-2 expressed on CD4-positive T cells to activate Th2 lymphocytes. High levels of L-PGDS increased the degree of tubular fibrosis, while L-PGDS-knockout mice and prostaglandin D2 receptor CRTH2-knockout mice revealed a reduction in renal fibrosis, which may be related to the reduced infiltration of Th2 lymphocytes as well as reduced generation of the Th2 cytokines IL-4 and IL-13 [85]. In contrast, the PGD_2_ metabolite, 15d-PGJ2, can activate PPARγ and modulate the adhesion process via inhibiting the TNFα-triggered IKK-NFκB pathway, which eventually suppresses inflammation in mouse renal tubular epithelial cells [86]. Moreover, PGs are also regulators of renal ischemia and vasoconstriction. Conversely, PGl_2_ and PGE_1_ have the effect of relaxing blood vessels and preventing hypoxia-mediated renal tissue damage. More importantly, in the case of renal artery stenosis, PGl_2_ and PGE_1_ selectively prevent tissue contraction and inhibit the decline of GFR [87,88].

It is well known that PGE_2_ is the major product of the COX-2 pathway in the kidneys, which mediates kidney damage [89]. PGE2 play a crucial role in renal hemodynamics, renin release, and renal tubular sodium/water resorption [90,91]. In pathological environments such as diabetic nephropathy, PGE2 synthesis is increased, which may affect cell proliferation, differentiation or apoptosis [92,93]. Current studies indicate that four different EP receptors for PGE_2_ are closely associated with rapidly progressive glomerulonephritis (RPGN) [94]. In humans, EP1 receptor mRNA is predominantly expressed in the glomerulus [95]. In diabetic mouse models, activation of EP1 promotes the progression of diabetic nephropathy [96]. On the other hand, deletion of EP1 inhibited the down-regulation of nephrin, and improved glomerular basement membrane thickening and foot process regression [96]. In contrast, a few studies have shown that EP1 deficiency caused severe renal impairment in mice with glomerulonephritis, and the cause of this difference has not yet been discovered [97]. Interestingly, PGE_2_ has both vasodilation and vasoconstriction effects on renal afferent arterioles. Treatment with high concentrations of PGE_2_ (between 1 and 10 nmol/L) caused arterial vasodilation, and at lower concentrations (0.100 nmol/L), PGE_2_ leads to vasoconstriction and aggravates hydronephrosis, which may be due to the binding of the EP3 receptor to the pertussis toxin (PTX)-sensitive G protein Gα_i_. [98]. Glomerular hypertrophy caused by unilateral nephrectomy in mice may be associated with increased expression of EP2 [99]. In netrin-1-deficient mice, proximal tubular injury can be attributed to the COX-2-PGE_2_-mediated inflammatory response [100]. Further studies have confirmed that activation of the COX-2-PGE_2_-EP2 axis may be a specific response of podocytes to fluid flow shear stress. This appears to provide a mechanistic basis for changes in podocyte structure and glomerular filtration barrier, leading to proteinuria in high filtration-mediated renal injury [99]. EP3 also plays a paramount role in the progression of renal diseases. *Ep3*^−/−^ -STZ mice have less volume of urine and higher urine osmotic pressure compared with wild-type STZ (WT-STZ) mice, indicating enhanced water reabsorption. In parallel, the expressions of aquaporin-1, aquaporin-2, and urea transporter A1 in *Ep3*^−/−^ -STZ mice were increased, so the presence of EP3 was an important promoting factor in the progression of renal disease [101]. EP4 is abundant in almost all types of kidney cells [95], and plays different roles at the glomeruli [102] and tubules [103]. The Gα_s_-conjugated EP4 receptor directly activates adenylate cyclase, increases cAMP, and also activates phosphoinositide kinase 3 [94,104]. In addition, EP4 also stimulates AMP-activated protein kinase and COX2 in mouse podocytes in a p38-dependent manner [105]. When the rats were treated with a low-salt diet, EP4 transcripts in the glomeruli increased significantly, which in turn mediated PGE_2_-induced renin secretion and maintained renal blood flow [106]. More importantly, the overexpression of EP4 contributed to podocyte injury and compromised the glomerular filtration barrier in podocyte-specific EP4 receptor transgenic (EP4^pod+^) mice after 5/6 nephrectomy [107]. On the other hand, in a rat model of cisplatin-induced renal failure, the COX1-PGE2-EP4 axis plays more important roles than dose COX-2 in regulating renal epithelial regeneration [108]. The endogenous PGE_2_-EP4 system is involved in tubule-interstitial fibrosis in a mouse UUO model. EP4 knockout significantly augmented obstruction-induced histological alterations. The effects of EP4 agonist are controversial. Use of EP4-sepicific agonist down-regulated the expression of renal macrophage chemokines and pro-fibrinogen growth factors, and inhibited the progression of renal inflammation [103]. In the rat model of acute renal failure, EP4 agonist reduces serum creatinine level and improve survival rate [109]. Moreover, in chronic kidney failure both EP2 and EP4 receptors are shown to be equally important in preserving the progression of chronic kidney disease [109]. In addition, PGE2 can alter renal cell growth, matrix transformation, fibrosis, and apoptosis by activating the EP4 receptor [89]. However, in a STZ-induced diabetic mouse model, the level of cytokines (TNFα and IL-6) and chemokines (MCP-1 and IP-10) in urine of EP4 agonist-treated mice were remarkably higher than those of vehicle-treated diabetes mice, which aggravated glomerular sclerosis and renal tubular interstitial fibrosis [110]. The exact mechanisms of EP4 agonists for the treatment of different kidney diseases need to be further studied.

### 3.2. HETEs and Renal Inflammation

It has been argued that the regulatory effects of HETEs on platelet function by autocrine or paracrine play a crucial role in the renal inflammation. In view of renal inflammation, which is often associated with renal vascular disease, HETEs have a dual role of anti-thrombotic and pro-thrombotic effects. 5-HETE, 12-HETE and 15-HETE inhibited platelet PLA_2_ activity [111]. Platelets are more sensitive to ADP-induced aggregation, and mouse platelets disrupted by gene targeting 12-lipoxygenase (P-12LO^−/−^) of the 12-LOX gene exhibit selective hypersensitivity to ADP [112]. Current study found that in patients with type 2 diabetes, exogenous 12-HpETE activates platelet p38 mitogen-activated protein kinase (p38 MAPK), which in turn promotes the platelet activation in the oxidative stress-related pathophysiological states [113]. Under oxidative stress, a decrease in the glutathione peroxidase activity promotes the formation of 12-HpETE and further causes an increase in the phosphorylation level of p38 MAPK, which further activates platelets [113], exacerbating the kidney damage [114]. Therefore, HETE-induced renal inflammation can be achieved through the regulation of platelets.

In recent years, increasing literatures reveal the activation of peroxisome proliferator-activated receptors (PPARs) by HETEs. PPARs, with three isotypes (PPARα, PPARδ, and PPARγ), are all transcriptional factors for lipogenesis and mainly expressed in the adipocytes and immune cells that regulate the expression of a quantity of genes associated with renal inflammation [115,116,117]. In macrophages, high concentrations of HETEs activate PPARγ [118,119], the latter gives rise to the increasing expression of CD36, which induces the apoptosis of tubule-interstitial cells, and ultimately leads to the inflammation and fibrosis of renal tubules [120]. In addition, HETEs can also activate PPARα [121], which has an important effect on renal inflammation by participating in the regulation of lipid metabolism [122,123]. Moreover, the metabolite of 20-HETE, 20-COOH-AA, is an endogenous dual activator of PPARα and PPARγ, and its efficiency to activate PPARγ is twice than that of 20-HETE, revealing the potential modulatory effects of 20-COOH-AA on renal inflammation via activation of PPARs.

The roles of HETEs are also of interest in regulating the renal ion transport during renal inflammation. 20-HETE has a wide range of biological effects, including cell proliferation and angiogenesis, and is a relevant factor that regulates the renal function [124]. 20-HETE is widely synthesized in the renal tubules, including proximal tubules, thick ascending rings, small arteries of the renal cortex and the outer medulla. 20-HETE can fully bind to major vasoconstrictors, such as angiotensin II [125], adrenaline, and endothelin [126]. As early as 1991, Escalante had pointed out that both 20-HETE and 20-COOH-AA, acting similarly to furosemide, reduce the Na^+^ and K^+^ concentrations of medullary thick ascending limb of Henle (mTALH) cells, suggesting the potential impact of HETEs on renal blood flow, GFR and urinary sodium excretion rates [127]. CYP4A11 is a key enzyme in the synthesis of 20-HETE. Both CYP4A and CYP4F belong to the CYP4 gene family, which catalyzes the conversion of arachidonic acid to 20-HETE [128]. The expression of CYP4A protein and the production of 20-HETE were significantly higher in the renal cortex and outer medulla of Dahl S salt-sensitive rats that fed either a low-salt or high-salt diet, indicating that the increased levels of renal 20-HETE in Dahl S rats promoted sodium retention and the development of salt sensitive hypertension [129]. Normalization of the 20-HETE levels with fibrates or transfer of wild-type CYP4A genes in congenic Dahl S strains is of great significance for inhibiting the progression of proteinuria and renal injury [129,130,131]. In the kidney, especially in the renal cortex, CYP4A also increases pressure natriuresis in the Dahl salt-sensitive rat [132]. The amount of 20-HETE metabolites in the CYP450 pathway produced by the extra renal medullary microsomes of salt-sensitive (SS/Jr) rats was significantly lower than that of salt-resistant (SR/Jr) rats [133].

In contrast, pharmacological inhibition of 20-HETE production in the outer medulla of the kidney in Lewis rats renders salt-sensitive [134], which means 20-HETE is a powerful vasoconstrictor in the kidney [135,136]. In renal microvessels, 20-HETE can strongly inhibit the activity of Na^+^/K^+^-ATPase and is a highly effective vasoconstrictor [137]. Increased production of 20-HETE in the renal microcirculation will decrease glomerular capillary pressure and GFR [124,138]. More importantly, 20-HETE can also reduce the pressure of post-glomerular circulation and ultimately promote the development of hypertension [138]. Interestingly though, 20-HETE-mediated increases in renal vascular resistance also reduce stress during glomerular circulation, thereby preventing the occurrence of glomerular damage caused by hypertension [139]. Therefore, when the synthesis of 20-HETE in the kidney is disturbed, the self-regulation of renal blood flow and the renal tubule-glomerular feedback system will be affected, resulting in reduced ion transport, increasing the difficulty of blood pressure control, and eventually aggravating glomerulonephritis and renal tubular interstitial nephritis [124]. The effect of 20-HETE on vascular function causes an ischemia-reperfusion injury (IRI) change that prolongs vasoconstriction after reperfusion and increased I/R injury [140]. It is well known that IRI is the most common cause of AKI, and 20-HETE production is elevated after renal ischemia (I/R) [141,142,143]. Use of 20-HETE analogues at a lower dosage can reduce the elevated plasma creatinine levels after I/R injury. In addition, 20-HETE analogues preconditioning reduced the area of tubular epithelial necrosis after I/R injury [143]. Moreover, 20-HETE can also be used to attenuate IRI by increasing medullary oxygen cooperation because it increases medullary blood flow and inhibits the renal tubular sodium transport [143,144]. On the other hand, in the kidney, 20-HETE overexpression significantly aggravated cell damage caused by I/R damage, which was mediated by the activation of caspase-3 and partly by enhanced CYP4A-producing free radicals [140]. Thus, the relationship between 20-HETE and AKI may be further clarified. 20-HETE may also promote the progression of renal inflammation by mediating apoptosis [145,146]. Studies have proved that CYP4A and NADPH oxidase expression was up-regulated in glomeruli of diabetic OVE26 mice [145]. And hyperglycemia increases 20-HETE production and enhances 20-HETE-dependent ROS formation and apoptosis in the mouse podocytes and rat tubuli epithelial cells, while 20-HETE blockade reduces ROS and improves apoptosis and albuminuria [145,146]. Recent research indicated that 20-HETE can increase the TRPC6 activity in podocytes, secondary to podocytes and activate ROS production [147]. Activation of TRPC6 results in the disappearance of the foot processes, but eliminates the detachment of podocyte [148]. Similar to 20-HETE, the correlation between 19-HETE and renal inflammation is mainly achieved through its effects on renal hemodynamics and blood pressure, which will not be repeated here [149].

### 3.3. LTs/LXs and Renal Inflammation

As the major metabolite of the LOX pathway, LTs and LXs both play considerable roles in kidney inflammation. LTs are a class of powerful chemotactic molecules that regulate leukocyte migration and activation [150]. Current studies have confirmed increases of LTs and LXs in the kidneys after ischemia, which further mediates a series of inflammatory reactions leading to the kidney damage [151]. More importantly, studies have shown that renal tissue can produce LTs without relying on circulating inflammatory cells [152]. The action of LTB_4_ on neutrophil aggregation and infiltration and kidney tissue damage in the rat iARF model was first reported. In the rat kidney ischemia-reperfusion (I/R) model, LTB_4_ played a leading role in the polymorphonuclear neutrophils (PMNs) [153]. LTB_4_-dependent cells migrate to ischemic renal parenchyma and activate neutrophils, which can rapidly up-regulate leukocyte adhesion molecules, and then promote the initial infiltration of PMNs. Activated neutrophils induce endothelial cell injury, and vasodilation is blocked, further aggravating ischemic tubulointerstitial damage and causing a vicious circle of renal tissue damage [153]. A recent study by Landgraf on LTB_4_-mediated renal tissue inflammation has found that LTB_4_ and LTD_4_ inhibited the endocytosis of albumin in LLC-PK1 cells (pig kidney cells), attenuating the activation of protein kinase C (PKC) and protein kinase B (PKB) and thereby reducing the absorption of albumin. Meanwhile, in mice models, LTs inhibited the secretion of the anti-inflammatory cytokine IL-10, hindered the PI-3K/PKB pathway, and caused albumin overload that finally gave rise to tubule-interstitial damage [154,155]. After a few minutes of reperfusion, PMNs were recruited and LTA_4_ was immediately transformed into LTB_4_ in the kidney. Once PMNs entered the interstitial space, the above mechanism will be performed at high speed, thus accelerating the extensive migration of PMNs and forming a vicious cycle of tissue damage [153]. The role of LTs in glomerular injury has also been demonstrated in the nephrotoxic serum-induced glomerular injury model, due to increased recruitment/activation of polymorphonuclear cells and increased LTB_4_ production in the kidney, which eventually amplifies the reduction of glomerular filtration rate [156]. Moreover, LTB_4_ and LTC_4_/D_4_/E_4_ levels are both increased in patients with nephrotic syndrome (NS), and this may be the reason why the serum creatinine, diastolic blood pressure, and protein/creatinine ratio of patients are significantly reduced [157].

As another major metabolite of the LOX pathway, LXs are important endogenous anti-inflammatory lipid transmitters that have been discovered earlier and can act on a variety of cells, including neutrophils, mononuclear macrophages, mesangial cells etc., to exert complex anti-inflammatory effects [158,159,160]. In glomerulonephritis, LXA_4_ inhibits LTs and reduces further infiltration of leukocytes [161,162]. In acute poststreptococcal glomerulonephritis (APSGN), one of the crucial pathological effects is the infiltration of neutrophils and monocytes in the glomeruli during the acute phase [163]. In vitro experiments with human mesangial cells showed that very low concentrations of LXA_4_ (1–10 nmol/L) inhibited the expression of cell fibrosis-related genes induced by platelet-derived growth factor and connective tissue growth factor [164]. Leukocytes infiltrate and LTB_4_ synthesis increases in rat kidneys at the early stage after nephrotoxic serum injection. At the same time, IL-4 and IL-13 produced by Th2 cells stimulate glomerular expression of LOX and synthesis of 15S-HETE and LXA_4_. The later inhibits LTB_4_ signaling, reduces leukocyte chemotaxis and transforms neutrophil infiltration into mononuclear macrophage infiltration, resulting in removal of apoptotic neutrophils and repairmen of tissue damage [165,166]. Another nephritis involving LXs is Henoch-Schonlein purpura nephritis (HSPN), which is a common secondary glomerulonephritis in pediatrics [167]. In HSPN, both plasma and urinary levels of LTB_4_ and LTE_4_ increase, while LXA_4_ levels decrease, indicating that endogenous LXA_4_ deficiency may be one of the causes of HSPN (which provides a basis for exploring new methods for the treatment of HSPN) [167]. Recent studies have shown that LXs inhibited the increase of renal/body weight ratio in diabetic animals and also reduced the glomerular dilatation and mesangial matrix deposition, in both high-fat diet-induced diabetes mouse model and unilateral ureteral obstruction (UUO) mouse model [168,169]. In addition, LXs can significantly reduce proteinuria and play an important role in reversing the CKD induced in diabetic *ApoE*^−/−^ male mice [170]. In cultured human renal epithelial cells, treatment with LXs reduced TNF-α-driven Egr-1 activation, which may regulate the inflammation of kidney disease [170] [171,172]. As Egr-1 activity is elevated in renal tubular cells in patients with renal failure [173], this may be related to Egr-1, which mediates the TGF-β signaling pathway in the kidney, immune cell infiltration, and regulates the NF-κB activity and cytokine/chemokine expression in the kidney [173]. Brennan et al. had identified that Egr-1 as a downstream target for LXs, besides, they also confirmed the interaction between LXs and Egr-1 in studies of human renal tubular epithelial cells [170]. In summary, LXs have a significant positive effect on inhibiting kidney inflammation.

### 3.4. EETs and Renal Inflammation

It has been reported that 14,15-EET can alleviate the proteinuria and renal dysfunction caused by cyclosporine, which may be related to the inhibition of inflammatory cells infiltration into the kidney and reduction of renal fibrosis [174]. In the rat model of renal tubule-interstitial inflammation induced by unilateral ureteral obstruction (UUO), sEH deficiency has a beneficial effect on renal fibrosis and interstitial inflammation. The molecular mechanism may be associated with lack of sEH and the reduction of EETs degradation, which inhibited the transforming growth factor (TGF)-1/Smad3 signaling, diminished infiltration of neutrophils and macrophages, prevented expression levels of NF-κB target gene proteins (TNF-α and ICAM-1), decrease cell death caused by ROS, and causes PPAR inactivation [175]. Manhiani proved that soluble epoxide hydrolase gene deletion attenuated renal injury and inflammation with DOCA-salt hypertension [176]. In the kidney, EET has anti-inflammatory effects by blocking the activation of NF-kB and inhibiting the progression of renal inflammation by reducing renal macrophage infiltration [176]. Research also showed that, increased production of EET prevented microalbuminuria and kidney inflammation in the hyperglycemic overweight mice [177]. However, in the kidney, EET/DHET-ratios were increased in sEH knockout mice, but surprisingly, plasma creatinine concentration and IRI were higher than in the control group, which may be due to the formation of 20-HETE that eliminates the potentially beneficial effects of EET degradation [178]. In an I/R-induced AKI mouse model, the administration of 14,15-EET alleviated the dilated renal tubules, leading to an obvious reduction in plasma Cr, TNF-α and IL-6 [179]. These effects, discovered in I/R-caused AKI mice, may be due to the 14,15-EET reversing the I/R-induced declination of p-GSK3β expression, which induced the ratio of p-GSK3β/GSK3β back to a normal level [180]. In addition, the CYP450-derived eicosanoids could activate the eNOS and NO release. 14,15-EET can activate the endothelial cell NO synthase (eNOS) and NO release by blocking the Ca^2+^-activated K^+^, which eventually caused afferent arteries dilatation and reduces renal inflammation [181]. 5,6-EET, found in the kidneys of rabbits, is involved in the metabolism of two types of vasodilators, one being PGE_2_/PGI_2_ and the other being the adenosine analogue 5,6-epoxy-PGE [182]. In contrast, in the rat I/R kidney model, 5,6-EET caused COX-dependent renal vasoconstriction, whereas in isolated rat kidneys, 5,6-EET dilated blood vessels. In spontaneously hypertensive rats, 5,6-EET and 11,12-EET induced renal vasodilation more than 2 times greater than in Wistar Kyoto rats [183]. Stimulation of EETs activating adenosine 2A (A2A) may be an important mechanism for regulating microvascular tension in glomeruli [184]. It has been reported that 11,12-EET may represent an A2A-mediated mediator of glomerular microvascular expansion in rats. In rat glomerular pre-microvasculature, EET release was an important step in the activation of A2A receptors and adenylyl cyclase activation, and EETs mediated the activation of the Gs alpha protein by stimulating mono-ADP-ribosyltransferase [185]. The mechanism by which 11,12-EET dilates the afferent arterioles is possibly due to the phosphorylation of protein phosphatase 2A activity and Ca^2+^-activated K^+^ channels [186]. Recent research has proven that 11,12-EET is a major product of Cyp2c44 in mice, which is involved in regulating the excretion of epithelial sodium in the collecting duct. Cyp2c44 in the collecting duct can promote the excretion of Na^+^ in the kidney under high salt or high K^+^ environment by inhibiting epithelial Na^+^ channel (ENaC), thus preventing excessive Na^+^ absorption, suggesting that 11,12-EET may indirectly affect the renal inflammation through its role in regulation of blood pressure [187]. It is worth mentioning that 8,9-EET has a unique protective effect on the glomerulus [188]. Exogenous 8,9-EET (1–1000 nM) dose-dependently prevented a circulating permeability factor (FSPF)-induced increase in the glomerular albumin permeability [188]. The other three EET regioisomers, 8,9-EET metabolite, 8,9-dihydroxyeicosatrienoic acid and unrelated 11,14-eicosadienoic acid were ineffective, indicating the specificity of 8,9-EET for glomerular protection [188]. More importantly, a synthetic analog of 8,9-EET containing a double bond antagonized the effect of 8,9-EET on FSPF-induced increase in glomerular albumin permeability. These results indicate that the development of stable analogs of 8,9-EET may make sense to the effective management of glomerular dysfunction [188].

## 4. Treatments

A number of drugs for kidney inflammation based on the AA metabolic pathway are in the early stages of development for human disease treatment, and their study output is limited. The treatment of kidney disease based on AA is varied and many factors are involved, so here we only introduce strategies that are relevant to humans (Table 1).

### 4.1. Phospholipase-Associated Therapy

Current research has suggested that the M-type phospholipase A2 receptor (PLA_2_R) is sensitive and specific for idiopathic membranous nephropathy (MN). Serum phospholipase A_2_ receptor antibody (PLA_2_R Ab) and circulating anti-phospholipase A_2_ receptor antibodies (anti-PLA2R Abs) are now regarded as a valuable indicator of prognosis for patients with nephritis, especially in patients with idiopathic membranous nephropathy (IMN) [203,204]. IMN is one of the most common causes of adult primary NS. The KDIGO guidelines have recommend treatment with glucocorticoids and immunosuppressive agents for IMN [191,194]. Beck et al. reported that 35 patients with IMN were treated with rituximab (the anti- autoimmune disease drug), within whom 71% of patients were serum anti-PLA_2_R -positive. After 12 months of rituximab therapy, serum anti-PLA_2_R decreased or disappeared in 17 (68%) of these patients. After 12 months and 24 months of therapy, the rates of complete remission (CR) and partial response (PR) were 59% and 88%, respectively, and the decrease in antibody titer was earlier than the remission of proteinuria [189]. An earlier study of 37 biopsy confirmed IMN patients indicated that, after receiving standard immunosuppressive therapy (cyclosporine combined with methylprednisolone), the titer of PLA_2_R-Ab positive patients gradually decreased with an ameliorated proteinuria [190]. A retrospective study that includes 113 IMN patients showed that, the tacrolimus (TAC) and corticosteroids, corticosteroids and cyclophosphamide (CYC) and corticosteroids alone respectively can decrease the serum PLA_2_R-Ab titer and proteinuria in IMN patients [192]. As a conventional anti-tumor drug, rituximab seems to have a favorable result on MN [205]. 8 months after receiving rituximab, 22 patients with PLA_2_R-related MN showed a decrease of proteinuria and PLA_2_R Ab titer. At the meantime, renal function remained stable, and serum albumin increased [205]. Similar results were obtained in Beck’s study, which suggested that rituximab can inhibit the progression of IMN by lowering the PLA_2_R Ab titer [189]. Lowered PLA_2_R Ab titer could also be achieved by oral administration of prednisolone or cyclical cyclophosphamide (CTX) or cyclophosphamide or mycophenolate mofetil (MMF) in combination with steroids in IMN patients, showing a beneficial result with elevated glomerular filtration rate (eGFR) [191,193]. A recent study showed that combination with corticosteroids and rituximab also decreases the serum creatinine and PLA_2_R-Ab level [206]. However, one cannot ignore the fact that steroid treatment aggravated tubule-interstitial fibrosis in patients with acute interstitial nephritis [207]. A recent study also reported that corticosteroid treatment seems to increase the recurrence rate of TIN [208], suggesting that the PLA_2_R-Ab therapy may only work in IMN patients. The data from this research seems contradictory, especially the small sample size, which makes it difficult to draw firm conclusions regarding their outcomes. Therefore, the prospect of corticosteroids for the anti- PLA_2_R-Ab therapy needs more research and exploration.

### 4.2. COX-Associated Therapy

Nonsteroidal anti-inflammatory drugs (NSAIDs) are the main COX inhibitors. Their common mechanism of action is the inhibition of COX, and the most important result of this inhibition is the reduction of the production of PGs, thus playing anti-inflammatory, pain relieving and antipyretic roles [209]. COX inhibitors fall into two broad categories: non-specific COX inhibitors and specific COX-2 inhibitors. Among them, aspirin, ibuprofen, naproxen, etc. are usually used as non-specific COX inhibitors in the treatment of nephritis. Certain functional groups of COX-2 inhibitors (coxibs) can insert into the hydrophobic cavity formed by some amino acid residues of COX-2, causing the loss of their catalytic function [18]. In this case, AA cannot perform biological transformations under the catalysis of COX-2, thus blocking the synthesis of PGs as well as the inflammatory process [58]. Aspirin, also known as acetylsalicylic acid, is a well-known antipyretic analgesic that inhibits platelet aggregation and prevents thrombosis, which could also be used in kidney inflammatory diseases [210]. Aspirin is a non-specific COX inhibitor that inhibits both COX-1 and COX-2, which has a good effect on the prevention of primary and secondary thrombosis [211,212,213]. Aspirin treatment in MRL/MpJ-*Fas^lpr^*/J (MRL/lpr) mouse could alleviate LN [195] and reduce the risk of platelet aggregation and micro-embolization, which may improve the GFR during renal perfusion, thereby improving kidney function [196]. At the same time, perioperative aspirin can reduce the thromboxane level in the urine, which is a powerful vasoconstrictor, and improve renal function [196]. A large prospective cohort study of more than 5000 patients undergoing cardiac surgery indicated that patients taking aspirin had a lower incidence of AKI than patients who did not take aspirin (*P* < 0.001) [214]. Above all, the incidence of renal failure was reduced by 74% in those who were taking aspirin [214]. However, according to a recent study, low-dose aspirin oral intake was shown to have no significant effect on kidney function during the 15 years after kidney transplantation [215]. This may be because aspirin can be rapidly hydrolyzed to salicylate immediately, a product that has an almost negligible effect on COX [210]. Further, the results found that aspirin increased the risk of massive hemorrhage and further increased the risk of subsequent AKI [197]. Since the findings in numerous studies are contradictory, additional research on the value of aspirin for kidney inflammation is essential.

As the most widely used NSAID, ibuprofen is the first choice for the treatment of inflammatory pain [198]. Compared with other NSAIDs and coxibs, ibuprofen has fewer side effects on the gastrointestinal tract and a relatively low incidence of liver and kidney damage [216]. However, the amount of urine was obviously reduced after the first day of treatment with ibuprofen in premature infants, and the serum creatinine concentration was significantly increased on the third day of treatment [217]. There seems to be a latent relationship between ibuprofen and acute tubule-interstitial nephritis (ATIN), which is a major factor contributing to the acute renal insufficiency that must be kept in mind when it is used for treatment [199,218,219].

Nimesulide, another NSAID, has selective inhibition of COX-2 and qualitative inhibition of COX-1, and the principal effect of nimesulide on kidney inflammation mainly involves renal hemodynamics and electrolyte excretion. The use of nimesulide in healthy volunteers during long-term use of furosemide causes a brief and acute decline in renal hemodynamics and attenuated the natriuretic, kaliuretic and diuretic effects of furosemide [220]. Nimesulide reduced the plasma renin activity, aldosterone levels, and urinary PTE_2_ levels [220]. Meanwhile, diuretic-induced renin activity was attenuated by nimesulide. This suggests that nimesulide protects kidney function by allowing sodium and potassium retention [220]. However, it is interesting to remark that there was no significant change in serum creatinine and Tamm-Horsfall glycoprotein (THG) concentrations as well as no significant effect on GFR after nimesulide in 16 healthy human volunteers [221]. The weak effect of nimesulide on renal toxicity may suggest that it has no strong inhibitory effect on renal COX at the therapeutic dose, which is similar to the results from Ceserani et al., who showed that nimesulide did not dramatically reduced urinary PTE_2_ excretion in rats [222].

Several other NSAIDs also plays active role in the treatment of nephritis. Carprofen, an inhibitor of COX-mediated PG synthesis, increased the thick ascending limb of the loop of Henle at the tubular level and increased resorption of solute in the medullary segment of the upper extremity, thereby reducing sodium and chlorine excretion and potentially improving GFR or overall excretion of solute in the human kidney [200]. Diclofenac acid can reduce the recurrence survival rate and improve the survival rate of patients with renal cancer after surgery, which may be related to the fact that diclofenac acid inhibits the production of PGE_2_, which in turn inhibited the process by which PGE_2_ alters intracellular cyclic adenosine monophosphate levels to reduce the number and activity of natural killer cells [223,224,225].

However, as a COX-2 inhibitor, rofecoxib increases the risk of major cardiovascular events during treatment [226]. In addition, naproxen and celecoxib were related to the occurrence of AKI [227]. Studies have associated people who take naproxen, regardless of the dosage, with a higher risk of nephrotic syndrome and AIN [228]. Other NSAIDs may also have potential adverse effects on renal function [229], such as indomethacin, which can cause acute sodium retention in healthy adults and reduce GFR levels by inhibiting COX [230]. Studies have shown that live-donor nephrectomy patients that were treated with ketolic acid had significant increases of urinary albumin/creatinine ratio after 1 year and is an independent risk factor for reducing GFR (odds ratio 1.38) [231]. A study from *Lancet* indicated that NSAIDs, such as rofecoxib, celecoxib, ibuprofen, naproxen, and diclofenac, increase the risk of vascular events during treatment [232]. However, it has also been pointed out that all COX-2 inhibitors did not significantly promote renal events and arrhythmia events [233]. These findings provide insight for future research, and prospective clinical trials are needed to assess the curative effect and safety of NSAIDs for the treatment of renal inflammation.

### 4.3. LOX-Associated Therapy

The research related to LOX-associated therapy is mostly studied on experimental animals, and when it comes to human studies, zileuton is often mentioned. Zileuton blocks the conversion of AA to LTB_4_ by inhibiting the 5-LOX activity and is often used for the prevention of inflammation-related diseases and for cancer treatment [234]. However, whether zileuton could be used to treat renal inflammation in humans still needs to be further confirmed by clinical trials. Licofelone is a novel dual anti-inflammatory drug that inhibits both COX and 5-LOX. A study has indicated that licofelone improved inflammation in human mesangial cells (HMCs) exposed to IL-18 in a dose-dependent manner, through inhibiting COX-2 enzyme activity and reducing PGE_2_ release in HMCs [201]. Similarly, licofelone inhibited IL-18-induced 5-LOX enzyme activity and thereby reduces leukotriene release. In addition, licofelone blocked IL-18-induced phosphorylation of p38 proliferation protein kinase, inhibited the expression of monocyte chemoattractant protein 1 and interferon-γ. Licofelone also suppressed mesangial cell proliferation caused by IL-18 [201]. These results indicate that licofelone alleviates human glomerular inflammation by inhibiting IL-18-induced proinflammatory cytokine release and cell proliferation. A more exciting result was that licofelone inhibited IL-18-induced mesangial cell proliferation, and the results indicated that licofelone might be effective for the treatment of glomerulonephritis in children [201]. 2,3-diarylxanthones, dual inhibitors of COX and 5-LOX, are capable of preventing the production of LTB_4_ in human neutrophils as well as decreasing PGE_2_ production in human whole blood in a concentration-dependent manner [235]. According to the current results, the effect of lox-related drugs in regulating inflammation is still in the experimental study stage. Baicalein, a 12/15-LOX inhibitor, was demonstrated to prevent the elevation in renal 12-HETE production and reduce renal inflammation in strptozotocin-induced diabetic mice [202].

### 4.4. CYP450/sEH-Associated Therapy

There are two pharmacological approaches that have been used to chronically elevate endogenous levels of EETs in order to evaluate their renal and vascular protective effects in vivo. One approach is to increase the levels of EETs by inducing epoxygenases with fibric acid derivatives such as clofibrate, fenofibrate, and bezafibrate [236]. Fenofibrate has been shown to strongly induce renal protein expression of CYP2C23, a major CYP epoxygenase in the rat kidney, and increase the renal epoxygenase activity [236]. We have previously reported that a CYP450 inducer, gemfibrozil, has shown positive effects on CYP2C-related non-alcoholic steatotic hepatitis [237]. More importantly, the potential link between dyslipidemia and renal inflammation also reveals the potential values of fibric acid derivatives for the treatment of renal inflammation [238,239] among diabetic dyslipidemia patients [240,241,242]. However, some reports suggested that, the combined action of metamizole and gemfibrozil could synergistically affect the proximal tubule and increase the chances of renal damage [243]. Some studies have shown that the epoxyeicosatrienoic acid analog can effectively slow down the kidney damage associated with oxidative stress, inflammation, and endoplasmic reticulum stress [244,245]. For example, a new oral drug, PVPA, reduces the proteinuria and renal dysfunction caused by cyclosporine, inhibits the inflammatory cell infiltration in the kidney, and reduces the renal fibrosis [174]. Warfarin, an anticoagulant, is mainly metabolized by CYP2C [246,247,248], which can inhibit the proliferation of mesangial cells by interfering with the activation of Gas6, and plays an important role in the treatment of various human kidney diseases, such as acute and chronic glomerulonephritis and diabetic nephropathy [249].

Another approach to elevate EETs is to inhibit the conversion of EETs to their less active metabolites by soluble epoxide hydrolase (sEH) [250]. sEH plays a major role in several diseases, including hypertension, cardiac hypertrophy, arteriosclerosis [251]. Because of its possible role in cardiovascular and other diseases, sEH is being pursued as a pharmacological target, and potent small molecule inhibitors are available [252]. Such inhibitors, like UC1153 (AR9281) and GSK2256294, were taken to clinical trials for treatment of hypertension and chronic obstructive pulmonary disease respectively [252]. However, even with the promise of epoxygenase metabolites to protect the kidney and vasculature, further research in this area is necessary in view of the small number of trials on humans.

## 5. Conclusions

The present review aims to discuss the effect of AA metabolism on kidney inflammation, as well as to provide a theoretical basis for the treatment of kidney inflammation. AA metabolism and kidney inflammation are closely linked in multiple ways. Through a summary of previous studies, our conclusions help us to understand the effects of AA metabolism on the kidney in several ways and provide the therapeutical treatment for renal inflammation. However, the studies on the treatment of kidney inflammation based on AA metabolism require a large sample of randomized controlled trails to elucidate their efficacy.

## Figures and Tables

**Figure 1 ijms-20-03683-f001:**
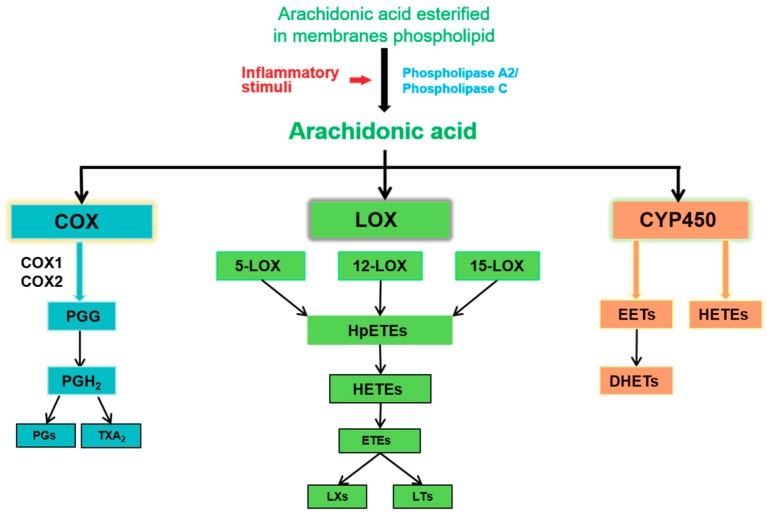
Scheme of eicosanoids biosynthesis pathways from arachidonic acid.

**Figure 2 ijms-20-03683-f002:**
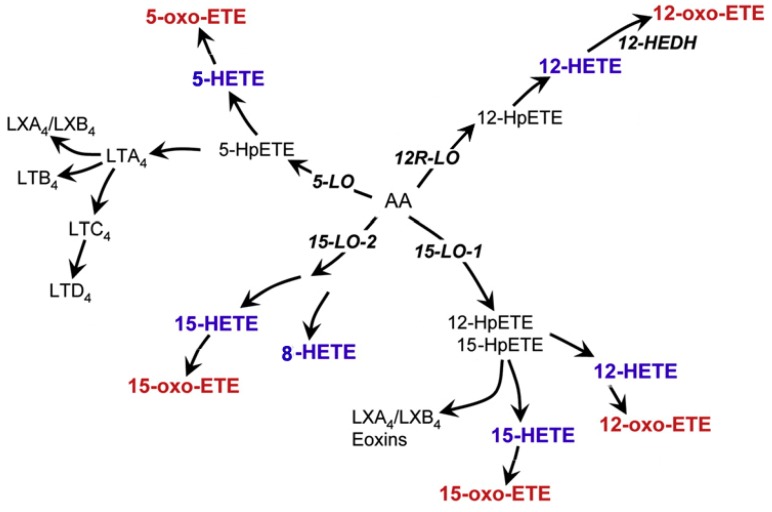
Lipoxygenase, and dehydrogenase pathways for the formation of HETEs, oxo-ETEs, and related eicosanoids.

**Table 1 ijms-20-03683-t001:** Drugs related to AA metabolism for kidney inflammation.

Compounds	Species	Targets	Kidney Disease	Outcome	Reference
Cyclosporine+ methylprednisolone	Human	Calcineurin and corticosteroid hormone receptor	IMN	Proteinuria ↓ PLA2R Ab ↓ Infiltration of defense cells ↓	[189]
Tacrolimus+corticosteroids, corticosteroids+cyclophosphamide, or corticosteroids alone.	Human	Peptidyl-prolyl isomerase and glucocorticoid receptors	IMN	Proteinuria ↓ Serum albumin ↑ Glomerular PLA2R ↓ Serum PLA2R-Ab ↓ Infiltration of defense cells ↑	[190]
Rituximab	Human	Pan-B-cell marker CD20	IMN/IgA Nephritis	Proteinuria ↓ GFR ↑ Serum albumin ↑ PLA2R Ab ↓ Infiltration of defense cells ↓	[191,192,193]
Prednisolone	Human	glucocorticoid receptors	IMN	GFR ↑ Proteinuria ↓ Serum albumin ↑ PLA2R Ab ↓ Infiltration of defense cells ↓	[194]
Cyclophosphamide or +corticosteroids	Human	glucocorticoid receptors (for corticosteroids)	IMN/IgA Nephritis	GFR ↑ Proteinuria ↓ Serum albumin ↑ PLA2R Ab ↓ Infiltration of defense cells ↓	[193,194]
Aspirin	Mouse and Human	COX-1/COX-2	AKI	GFR ↑ Serum creatinine ↓ Urinary output ↑ Proteinuria ↓	[195,196]
Ibuprofen	Human	COX-1/COX-2	ATIN	Pain control ↓	[197,198]
Nimesulide	Rats and Human	COX-1/COX-2	ATIN	Plasma renin activity ↓ Aldosterone level ↓ Urinary PTE2 level ↓	[199]
Indomethacin	Human	COX-1/COX-2	Renal failure	IL-6 ↓ IL-10 ↓	[199]
Carprofen	Human	COX-2	Renal failure	IL-1β ↓	[199]
Diclofenac acid	Rats and Human	COX1/COX2	Renal cancer	PGE2 level ↓	[200]
Zileuton	Human Mesangial Cells	LOX/COX-2	Renal cancer	Serum creatinine ↓ Interstitial fibrosis ↓	[201]
Licofelone	Mouse and Human	5-LOX/COX	Glomerulonephritis	IL-18 ↓ PGE2 ↓	[201]
Baicalein	Mouse	12/15-LOX	Diabetic nephropathy	12-HETE ↓ IL-6 ↓ Proteinuria ↓	[202]
PVPA	Rats	CYP450	Acute and chronic glomerulonephritis	Proteinuria ↓ Apoptosis in tubular epithelial cells ↓ Generation of reactive oxygen species ↓	[174]

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
