# Peer review of "Arachidonic Acid Metabolism and Kidney Inflammation"

_ijms, 2019, doi:10.3390/ijms20153683_

Round 1

Reviewer 1 Report

The authors have now significantly ameliorated the work regarding different aspects.

A minor point  relates to table 1- in the column "outcome", increased (↑) "infiltration of defense cells" is indicated as an outcome parameter. Shouldn't it be decreased (↓) ?

Author Response

The authors have now significantly ameliorated the work regarding different aspects.

A minor point  relates to table 1- in the column "outcome", increased () "infiltration of defense cells" is indicated as an outcome parameter. Shouldn't it be decreased () ?

answer:

Authors appreciate this correction. In table 1, column “outcome”, the outcome parameter “infiltration of defense cells” is indicated as decreased () now.

Reviewer 2 Report

Authors have addressed all my concerns. Review can be accepted in present form.

Author Response

English language and style

(x) Moderate English changes required 

answer:

Authors appreciate this suggestion.  English modification is done by a native English speaking colleague.

This manuscript is a resubmission of an earlier submission. The following is a list of the peer review reports and author responses from that submission.

Round 1

Reviewer 1 Report

The connection of arachidonic acid (AA) metabolism and kidney inflammation is of clinical value since the issue it may offers insight into positive effects and side effects of drugs interfering with (AA) metabolism. The authors present many biochemical informations. However, the data presented are very complexe and the base for treating kidney inflammation is a little bit unprecise, because “kidney inflammation” is completely different in divers kidney diseases. Table 1 is a little bit confusing. -Why is the target always mentioned with “PLA2 Rec AB” in the first six rows of table 1. PlA2Rec Abs is important in membranous glomerulonephritis , but not more generally. -Also the categories e.g. ” targets” in table 1 is unclear . PLA2R and cox-n are not comparable “targets”. In addition, Ibuprofen, Aspirin… are also “NSAIDS”. -In addition, many glomerulonephritis and interstitial nephritis forms are accompanied by infiltration of defense cells which play a key role in these disease. This s not adequately addressed. -The link of AA to important kidney diseases and potential implication of a therapy , such as IgA Nephritis ( one of the main kidney disease found histologically in adults) is not mentioned or described. -In table 1, in the rows labeled with “NSAIDS” , GFR amelioration is cited as outcome. However NSAIDs alter renal blood circulation and rather decrease GFR, this is a common cause of renal failure in the elderly with multipharmacy prescribed. - In sum, it is not clarified when NSAIDs (and what kind of) are favorable, and for which kidney disease.

Reviewer 2 Report

In this review, the authors have written extensively and very densely regarding Arachidonic Acid Metabolism and Kidney Inflammation. Review is very well written and authors have covered all respective points related AA metabolism. Review is very informative and intelligently written. However I have some concerns, after addressing these concerns review can be accepted.

Minor Points:

1.   Authors should provide more schematic illustration for the review. As such review appears to be more chatty and dense, hard to follow sometime. Couple of illustration will certainly help readers to follow the information more correctly.

2.   In the LOX pathway can author describe among ALOX-5, AlOX-8, ALOX-12, ALOX-15 which one is more important in renal injury (AKI or CKD).

3.   In Cytochrome P450 (CYP450) pathway paragraph. Authors should replace Confirmed with Confined in line which State CYP2J5 is an epoxygenase enzyme that is confirmed in tubular cells and metabolizes AA to 8,9-EET; 11,12-EET; and 14,15-EET in mouse kidney.

4.   Section 3.2- author should clearly review the ambiguous role of 20-HETE.

5.   Please replace what’s more in all section with some good scientific terminology, this word is not scientific at all.

6.   In table 1- author have mentioned compounds or inhibitors pertaining to most of the enzymes involved in AA metabolism. Is there any drug or inhibitor for ALOX-12 and ALOX-15 which is being used or researcher will tend to use in future for human.

7.   Section 4.1- An earlier study of 37 patients with biopsy confirmed IMN indicated. Please replace with indicated.